# FK506-Binding Protein 2 Participates in Proinsulin Folding

**DOI:** 10.3390/biom13010152

**Published:** 2023-01-11

**Authors:** Carolin Hoefner, Tenna Holgersen Bryde, Celina Pihl, Sylvia Naiga Tiedemann, Sophie Emilie Bresson, Hajira Ahmed Hotiana, Muhammad Saad Khilji, Theodore Dos Santos, Michele Puglia, Paola Pisano, Mariola Majewska, Julia Durzynska, Kristian Klindt, Justyna Klusek, Marcelo J. Perone, Robert Bucki, Per Mårten Hägglund, Pontus Emanuel Gourdon, Kamil Gotfryd, Edyta Urbaniak, Malgorzata Borowiak, Michael Wierer, Patrick Edward MacDonald, Thomas Mandrup-Poulsen, Michal Tomasz Marzec

**Affiliations:** 1Inflammation, Metabolism and Oxidation Section, Department of Biomedical Sciences, University of Copenhagen, 2200 Copenhagen, Denmark; 2Novo Nordisk Foundation Center for Basic Metabolic Research, University of Copenhagen, 2200 Copenhagen, Denmark; 3SAXOCON A/S, 2830 Virum, Denmark; 4Department of Molecular Medicine, Institute of Basic Medical Sciences, Faculty of Medicine, University of Oslo, 0316 Oslo, Norway; 5Department of Biomedical Sciences, University of Copenhagen, 2200 Copenhagen, Denmark; 6Department of Biology, Linderstrøm-Lang Center for Protein Science, University of Copenhagen, 2200 Copenhagen, Denmark; 7Department of Physiology, University of Veterinary and Animal Sciences, Lahore 54000, Punjab, Pakistan; 8Department of Pharmacology and Alberta Diabetes Institute, University of Alberta, Edmonton, AB T6G 2E1, Canada; 9Proteomics Research Infrastructure, University of Copenhagen, 2200 Copenhagen, Denmark; 10Department of Genetics, Institute of Experimental Biology, Faculty of Biology, Adam Mickiewicz University, 61-712 Poznań, Poland; 11Laboratory of Medical Genetics, Department of Surgical Medicine, Collegium Medicum, Jan Kochanowski University, 25-406 Kielce, Poland; 12Immuno-Endocrinology, Diabetes & Metabolism Laboratory, Instituto de Investigaciones en Medicina Traslacional, Facultad de Ciencias Biomédicas, CONICET-Universidad Austral, Buenos Aires B1629AHJ, Argentina; 13Department of Medical Microbiology and Nanobiomedical Engineering, Medical University of Białystok, 15-089 Białystok, Poland; 14Institute of Molecular Biology and Biotechnology, Faculty of Biology, Adam Mickiewicz University, 61-712 Poznań, Poland; 15Institute of Health Sciences, Collegium Medicum, Jan Kochanowski University, 25-406 Kielce, Poland

**Keywords:** FKBP2, proinsulin, proline isomerization, endoplasmic reticulum

## Abstract

Apart from chaperoning, disulfide bond formation, and downstream processing, the molecular sequence of proinsulin folding is not completely understood. Proinsulin requires proline isomerization for correct folding. Since FK506-binding protein 2 (FKBP2) is an ER-resident proline isomerase, we hypothesized that FKBP2 contributes to proinsulin folding. We found that FKBP2 co-immunoprecipitated with proinsulin and its chaperone GRP94 and that inhibition of FKBP2 expression increased proinsulin turnover with reduced intracellular proinsulin and insulin levels. This phenotype was accompanied by an increased proinsulin secretion and the formation of proinsulin high-molecular-weight complexes, a sign of proinsulin misfolding. FKBP2 knockout in pancreatic β-cells increased apoptosis without detectable up-regulation of ER stress response genes. Interestingly, FKBP2 mRNA was overexpressed in β-cells from pancreatic islets of T2D patients. Based on molecular modeling and an in vitro enzymatic assay, we suggest that proline at position 28 of the proinsulin B-chain (P28) is the substrate of FKBP2’s isomerization activity. We propose that this isomerization step catalyzed by FKBP2 is an essential sequence required for correct proinsulin folding.

## 1. Introduction

Misfolding of the insulin prohormone, proinsulin (PI), is increasingly appreciated as an early pathogenic event in type 2 diabetes (T2D) when a mismatch is reached between PI biosynthetic demand, typically increased due to insulin resistance, and folding capacity [1,2]. PI misfolding perturbs the insulin production machinery, causing decompensation of insulin secretion relative to need, and is believed to eventually contribute to a reduction of functional β-cell mass due to ER stress-induced apoptosis [1,2,3,4]. PI folding involves multiple steps, including chaperoning and disulfide bridge formation [5,6], all facilitating the locking of the linear amino acid chain into the 3D structure that, after further processing to mature insulin, lends the biological activity to the molecule.

The amino acid proline (Pro) adopts either a cis or trans isomeric conformation as it is incorporated into the polypeptide chain. The relatively low energy difference between cis and trans isomers of any amino acid (X) to Pro peptide bonds typically allows the presence of 3–10% cis X-Pro isomers in proteins, depending on the upstream amino acid. Nonetheless, usually only trans conformation is favorable for protein folding [7,8].

The transition of a proline residue from cis to trans conformation, or vice versa, is called cis–trans isomerization. This process occurs within the nascent amino acid chain and is a rate-limiting early step in protein folding that can trap a proline residue in an isomeric state, preventing proper folding [9,10]. Human PI contains three prolines, of which only the proline at position 28 (P28) in the B-chain is evolutionarily conserved (see below). The only publicly available crystal structure of folded PI contains an amino acid swap at the P28 position (P28K29 to K28P29), and molecule P29 is in trans conformation (Protein Data Bank structure [11]). During protein synthesis, the majority of P28 residues are expected to be trans-proline isomers and to fold readily. The remaining P28 residues will be cis-proline isomers and will require cis-to-trans isomerization to allow subsequent folding.

Proline isomerization is catalyzed by peptidyl-prolyl cis–trans isomerases [12,13,14]. Interestingly, broad pharmacological inhibition of these isomerases with, e.g., the potent immunosuppressant FK506, deteriorates insulin mRNA levels, glucose-stimulated insulin secretion, β-cell survival and leads to diabetes development in immunosuppressed patients post transplantation [15,16,17,18,19]. Recently, a P28 to leucine mutation has been reported in a case of maturity-onset diabetes of the young (MODY) presenting defects in PI folding, trafficking and dominant-negative behavior in vitro, emphasizing the clinical importance of cis–trans isomerization in insulin biosynthesis [20].

FK506 binding protein 2 (FKBP2) is one of seven proline isomerases expressed in the ER, where it partakes in protein folding and is induced in response to the buildup of unfolded proteins during ER stress [21,22]. Despite this, the substrates of FKBP2 are yet to be determined. Taken together, we hypothesized that P28 isomerization is an early event during PI folding executed by FKBP2.

Here, we show that FKBP2 is a novel PI binding partner in model β-cells and that P28 is a target for FKBP2-dependent isomerization. We demonstrate that FKBP2 binds to the recently described PI chaperone Glucose Regulated Protein 94 (GRP94), which is essential for PI handling [5]. Additionally, we observed that FKBP2 knockout (KO) in insulin-producing cells leads to loss of intracellular PI via increased misfolding, loss of PI solubility, and increased secretion of immature PI. Furthermore, while ER stress markers remain unchanged, FKBP2 KO results in increased cellular apoptosis. Finally, we show that FKBP2 mRNA is overexpressed in β-cells in human islets from organ donors with T2D patients, likely as a compensatory response.

## 2. Materials and Methods

### 2.1. Cell Culture

Rat insulinoma INS-1E cells (wild-type, FKBP2 control, and FKBP2 KO) were cultured in RPMI-1640 complete medium (for details, see Appendix A) at 37 °C with 5% CO_2_.

### 2.2. Generation of FKBP2 CRISPR/Cas-9 Mediated Knockout INS-1E Cell Lines

FKBP2 knockout INS-1E cells were generated with ready-to-use Lentiviral plasmids encoding a guide-RNA (gRNA) sequence targeting rat *fkbp2* exon 2 (Appendix A).

### 2.3. FKBP2 Expression in Pancreatic Endocrine Cells

The raw sequencing reads on human non-diabetic islets (deposited at EMBL-EBI under accession number E-MTAB-5061) and processed gene expression matrices along with the cell type inferred were reanalyzed searching for FKBP2 mRNA presence in human adult islets, including β-cells. Human pancreas sections were obtained from Collegium Medicum in Bydgoszcz or Nicolaus Copernicus University in Torun (Poland) under protocols approved by the Bioethical Commission (KB381/2020). Human pancreata were dissected from 3 non-diabetic donors: 45-year-old male (BMI 22.3, A1c 4.9%), 59-year-old female (BMI 35.2, A1c 6.0%, this donor’s characteristics might be considered prediabetic) and 62-year-old female (BMI 29.6, A1c 5.2%). The pancreas was fixed in 10% neutral buffered formalin for 24 h (Alpinus Chemia, Poland) and further processed using the automated histo-processor (Excelsior AS; Thermo Fisher Scientific, Poland), according to routine protocol. The 4 μm sections of adhered glass slides (SuperFrost plus, Bionovo, Poland) were deparaffinized and subjected to a citrate buffer antigen retrieval protocol before proceeding to immunofluorescence staining (IF). Three to five sections per donor were then incubated with primary antibodies diluted in NDS overnight at 4 °C. Primary antibodies used in the study are listed in Appendix A. After two 10 min washes with 0.1% Tween-20 in phosphate-buffered saline (PBS), cells were incubated with secondary antibodies conjugated to AlexaFluor488, TRITC or AlexaFluor647 (all Jackson Immunoresearch, USA) diluted 1:800 in NDS for 2 h at room temperature. Secondary antibodies were then washed out twice for 5 min with PBS-Tween-20 and incubated with DAPI (Sigma Aldrich, Poland) as a counterstain. Tissue samples were mounted with ProLong Diamond Antifade (Thermo Fisher Scientific, Poland) prior to imaging with Nikon Confocal and Leica inverted microscope DMIL LED. Adjacent tissue sections were stained with hematoxylin-eosin and evaluated by a pathologist for the absence of abnormal cellular phenotypes.

### 2.4. Immunoprecipitations and Mass Spectrometry Analysis

INS-1E cells were transfected at 60% confluence using Lipofectamine™ 3000 (ThermoFisher Scientific, Denmark) with plasmids coding for GFP-tagged GRP94, PI, ER-localized GFP [23,24] or myc-tagged FKBP2, according to the manufacturer’s protocol. Proteins were immunoprecipitated (IP) using the magnetic GFP- or myc-trap beads (Chromotek, Germany). IP samples were reduced and alkylated, digested with trypsin/LysC and the resulting peptides were analyzed on a Bruker Impact II ESI-QTOF (Bruker Daltonics, USA) mass spectrometer (Appendix A).

### 2.5. Size Exclusion Chromatography

Whole-cell lysates were fractionated by size exclusion chromatography (SEC) using a SuperdexTM 75 10/300 GL column (SigmaAldrich, Denmark) in Tris buffer (50 mM Tris pH 8.0, 150 mM NaCl, 5 mM KCl). Fractions of interest were analyzed via SDS-PAGE and immunoblotting (Appendix A).

### 2.6. Immunoblotting

Proteins were detected in cellular lysates, soluble, and non-soluble (pellets) fractions, and cell culture supernatants using immunoblotting with specific antibodies (Appendix A) under non-reducing and reducing conditions (Appendix A).

### 2.7. Single-Cell RNA Sequencing

Published datasets of single-cell RNA sequencing (scRNAseq) from dual patch-clamp sequencing studies were used to evaluate FKBP2 expression [25,26]. Cell types were assigned by Uniform Manifold Approximation and Projection with Leiden Clustering, and the expression of insulin, glucagon, pancreatic polypeptide, or somatostatin was analyzed.

### 2.8. In Silico Modeling of Protein Interactions and Protein Sequences Analysis

Using the ZDOCK 3.0.2 modeling software [27], crystal structures from the Protein Data Bank [28] of FKBP2 (PDB ID: 2PBC [29]) and PI (PDB ID: 2KQP, an NMR structure [11]) were used to generate prediction models of their interaction. Using the Python script ‘InterfaceResidues’ [30], residues predicted to interact were identified in PyMOL (Schrödinger, USA) using a 2.5 Å vicinity as a cut-off. FKBP2 evolutionary conserved residues were identified based on the literature [31] and alignment of protein sequences among FKBP family members. FKBP2 promotor analysis was performed within 500 bp upstream of the FKBP2 start site using TRANSFAC 2.0. Insulin 1/2 and IGF 1/2 amino acid sequences were retrieved from UniProtKB v. 2021_04 and aligned with BlastP.

### 2.9. FKBP2 Protein Activity Assay

n-Nitroanilide modified peptide (GERGFFYTPF-F-pNA, GenScript, USA) was mixed with assay buffer (DPBS pH 7.4), recombinant FKBP2 (1 µg, Abcam, Great Britain) or assay buffer alone and/or chymotrypsin (20 mM, Roche, Switzerland) to initiate the reaction. The increase in absorbance (405 nm) was read in 1 min intervals for 12 h at 25 °C. The signal from the time point with the highest absorbance in the reaction that contained FKBP2, n-Nitroanilide modified peptide, and chymotrypsin was used to compare experimental conditions and generate the graph (Appendix A).

### 2.10. qRT-PCR

The relative mRNA level of ER stress markers and insulin genes was determined by quantitative RT-PCR using specific primers (Appendix A; Appendix A).

### 2.11. Apoptosis Assay

Apoptotic cell death was determined by the detection of DNA–histone complexes present in the cytoplasmic fraction of cells using the Cell Death Detection ELISAPLUS kit (Roche, Switzerland) according to the manufacturer’s protocol (Appendix A).

### 2.12. Glucose-Stimulated Insulin-Secretion (GSIS)

INS-1E cell lines (control and FKBP2 KO) were examined for PI and insulin secretion in response to 2 and 20 mM glucose according to standard protocols (Appendix A).

### 2.13. Statistics

Differences between the two groups were assessed by two-tailed Student’s t-test or by one-way ANOVA of treatments versus control using the GraphPad Prism 9 (La Jolla, USA). Data are presented as means± SD. *p*-values ≤ 0.05 were considered significant.

## 3. Results

### 3.1. FKBP2 Directly Interacts with Proinsulin, and FKBP2 mRNA Is Overexpressed in Islets from Type 2 Diabetes Patients

To identify new PI interacting partners with the potential to impact PI folding, we exogenously expressed GFP-tagged PI and GRP94 (a novel PI chaperone [5]) in INS-1E cells. The GFP-tag enabled us to immunoprecipitate substantial amounts of target proteins for the detection and identification of co-immunoprecipitated partners during subsequent mass spectrometry analysis (Figure 1A). Among proteins localizing to the ER lumen, 11 were found to bind both PI and GRP94 (Figure 1A), including FKBP2 (Appendix A). Western blotting (WB) analysis confirmed the co-precipitation of myc-tagged FKBP2 to PI and GRP94 (Figure 1A).

The murine FKBP2 promoter contains regulatory elements characteristic for genes induced during the unfolded protein response (UPR) [22], and analysis of the 5′ flanking region of human FKBP2 identified similar regulatory elements, e.g., SP1, ATF3/6, and three incomplete ER stress response element sites (ERSE, Figure 1B), suggesting a role for FKBP2 in maintaining ER homeostasis.

Single-cell FKBP2 gene expression analysis revealed FKBP2 mRNA in all pancreatic islet cell types (Figure 1C, data reanalysis from [32]), and FKBP2 protein expression was confirmed in human β-cells (Figure 1D). As the increased demand for insulin in T2D may require compensatory up-regulation of chaperones involved in PI folding, we expanded the analysis to T2D cases. The results showed a significant up-regulation of FKBP2 in β-cells of T2D donors (Figure 1E, [25,26]); however, these data have not been independently validated (e.g., by qPCR).

### 3.2. Proinsulin P28 Is a Potential Target of FKBP2-Dependent Isomerization

Through ZDOCK modeling software and structures of FKBP2 (PDB ID: 2PBC) and PI (PDB ID: 2KQP), their interaction was modeled, demonstrating that PI fits into the FKBP2 substrate binding pocket (Figure 2A,B). Of note, FKBP2 exists mainly as a monomer in the β-cell, as evidenced by size exclusion chromatography (SEC) analysis (Appendix A). The majority of PI residues interacting with FKBP2 are located in the PI B-chain (Figure 2C) and include P28. On FKBP2′s side of the interaction, almost all evolutionary conserved residues are presumed to facilitate FKBP2′s isomerase activity interaction with PI (Figure 2D). The PI 2KQP NMR structure (Figure 2B,C) represents PI with an introduced mutation swapping the P28-K29 positions. Despite this, the in silico analysis identified P28 (P29 in the crystal structure) as the only plausible site of FKBP2-dependent proline isomerization. Furthermore, the P28 residue is fully evolutionary conserved within PI and the insulin-like growth factor family of hormones (Figure 2E,F), underscoring its structural and/or functional significance.

Furthermore, we set up a proline cis–trans isomerization assay, where peptide-containing PI B chain residues 20–28 were modified to have the last residue, P28, followed by phenylalanine (F) linked to the fluorophore paranitroanilin (pNA). The peptide bond between P28-F-pNA can be proteolyzed by chymotrypsin only when proline is in trans conformation [33]. The peptide was synthesized using cis and trans prolines and thus should contain a mixture of both isomers. The highest absorbance, representing free fluorophore pNA, was observed when FKBP2 was added to the reaction (Figure 2G), indicating that FKBP2 increased the presence of trans prolines in the reaction mixture.

### 3.3. Diminished Intracellular Proinsulin and Insulin Contents after FKBP2 KO

We performed CRISPR/Cas9-induced KO to evaluate the effects of FKBP2 deficiency in INS-1E cells on PI and intracellular insulin levels (Appendix A). FKBP2 KO cells cultured for 2 h in 20 or 2 mM glucose-containing media showed a significant reduction in intracellular PI contents (40% compared to control) as evidenced by WB analysis (Figure 3A). Similarly, insulin levels were reduced (40%) at 20 mM glucose conditions. We observed no significant change in insulin levels at 2 mM conditions (Figure 3A). Reconstitution of FKBP2 expression led to an increase in intracellular PI levels but had no effect on insulin contents at 20 mM glucose (Figure 3B).

Mutations resulting in the swap of PI residues P28 and K29 create conformational alterations that inhibit or weaken the formation of insulin dimers, allowing for more rapid absorption of fast-acting pharmaceutical insulins, e.g., LisPro [34,35]. Moreover, the amino acid sequence of pro-IGF-1, which does not naturally dimerize (but undergoes oligomerization [36]), resembles that of non-dimerizing insulins (Figure 2F). Considering this along with the FKBP2-PI modeling indicating P28 as a site of isomerization, we performed SEC separation of intracellular PI based on molecular mass. We detected no difference in the presence of PI monomers and dimers in FKBP2 KO cells when compared to the controls (Figure 3C), indicating that potential FKBP2-dependent P28 isomerization does not influence PI dimer formation.

### 3.4. FKBP2 KO Does Not Lead to ER Stress but Induces β-Cell Apoptosis

We then examined the activation of the UPR pathways in FKBP2 KO cells. We investigated mRNA expression levels of BiP, PERK, CHOP, ATF6, IRE1, and unspliced and spliced levels of XBP-1 and observed no statistically significant up-regulation in FKBP2 KO cells when compared to controls. Eighteen hours of treatment with 0.7 µM thapsigargin served as a positive control and induced a robust ER stress response (Figure 4A). ER stress has been reported to induce INS-1/2 mRNA degradation by activating the endonuclease IRE-1 [37,38,39], yet the results indicated that Ins-1/2 mRNA levels were not significantly reduced in FKBP2 KO cells (Figure 4A), supporting our conclusion that this ER stress-activated pathway is not triggered. We reasoned that FKBP2 KO and the loss of intracellular PI might induce cell apoptosis by means independent of ER stress (discussed below). Using a DNA–histone complex detection assay, we observed a significant increase in apoptosis in FKBP2 KO cells compared to controls, which was further up-regulated upon thapsigargin treatment (Figure 4B).

### 3.5. Proinsulin Turnover Is Increased in FKBP2 KO Cells Accompanied by Increased Proinsulin Secretion

As INS-1/2 mRNA levels were comparable between control and FKBP2 KO cells (Figure 4A), but steady-state PI levels were lower in FKBP2 KO cells, we reasoned that PI degradation at the ER stage or in the post-ER compartments during FKBP2 deficiency results in increased intracellular PI protein turnover. Consistently, following inhibition of protein synthesis with 100 µM cycloheximide (CHX), cellular PI levels were reduced by 50% after 2 h in FKBP2 KO lysates, whereas only a 20% reduction was observed after 4 h in control cell lysates (Figure 5A, B), demonstrating shorter PI protein half-life in FKBP2 KO cells. Brefeldin A (BFA) treatment has been shown to efficiently block ER to Golgi protein translocation and thereby inhibit PI secretion [6]. To determine if the shorter half-life was caused by an increased translocation and secretion, we pre-treated cells with 200 nM of BFA, inhibiting ER–Golgi anterograde transport before cycloheximide exposure. BFA did not change PI half-life significantly between Ctrl and Ctrl + BFA or FKBP2 KO and FKBP2 KO + BFA. However, when those changes were analyzed between pooled Ctrl (CHX and CHX + BFA) and FKBP2 KO (CHX and CHX + BFA) conditions, the average change of PI content over the time course of the experiment was reduced in the Ctrl group and increased in the FKBP2 KO, and the pattern was statistically significant (Figure 5B bottom). Taken together, this result could indicate that our observations are the result of a combined moderate ER stress caused by BFA (lowering PI content) and primary inhibition of PI secretion (increase in PI content) and suggests that PI secretion plays a role in shortening PI half-life in the FKBP2 KO phenotype.

To further examine this, we analyzed PI secretion during a 4 h 20 mM glucose stimulation and found that FKBP2 KO cells secrete significantly more PI than control cells, with this difference being less pronounced at 2 mM glucose stimulation (Figure 5C), further suggesting that PI secretion is at least partially responsible for the observed reduction in intracellular PI levels.

Finally, we expected that FKBP2 KO-related changes in PI folding would lead to lower secretion of insulin during glucose-stimulated insulin secretion (GSIS). However, that was not the case, as no observable difference in mature insulin secretion in response to glucose stimulation was reported when comparing control and FKBP2 KO cells (Figure 5D). Interestingly, the same experimental supernatants showed increased PI secretion at 2 and 20 mM glucose conditions (Figure 5D), aligning with the results from accumulated PI secretion (Figure 5C).

### 3.6. FKBP2 KO Induces Proinsulin Misfolding and Decreases Its Solubility

As FKBP2 KO induces intracellular PI loss, we wanted to identify the stage of the PI synthesis involving FKBP2. So far, disulfide bond formation is the only discrete step in PI folding described [40]. To evaluate if FKBP2 preferentially binds to unfolded and reduced PI or oxidized and folded PI, we pretreated INS-1E cell lysates for 10 min with 100 mM of reducing agent 2-mercaptoethanol (2-ME) prior to FKBP2 immunoprecipitation. We observed a substantial increase in FKBP2 binding to reduced PI, i.e., before disulfide bond bridges are formed (Figure 6A), suggesting that PI-FKBP2 interaction occurs at an early PI folding stage. Furthermore, we evaluated the presence of PI high-molecular-weight complexes (PI multimers), as they represent faulty disulfide bond formation and are indicative of PI misfolding ([1] and Appendix A). When analyzed by non-reducing SDS-PAGE, control and FKBP2 KO cells showed similar levels of PI multimers at 2 and 20 mM glucose concentrations (Figure 6B). However, the proportion of monomers/multimers in FKBP2 KO was significantly decreased, due to lower levels of PI monomers detected after treatment with 20 mM glucose.

We reasoned that some PI multimers in FKBP2 KO cells reach a state of lower solubility and consequently are not detectable in soluble cell fractions, resembling insulin-like growth factor 2 oligomerization that can transition into irreversible aggregates [36]. We, therefore, separated non-reduced control and FKBP2 KO cell lysates via SEC and analyzed its void fraction of protein aggregates. Upon SDS-PAGE and WB analysis, FKBP2 KO cells demonstrated a threefold increase in PI complexes as compared to controls (Figure 6C). Finally, SDS-PAGE and WB analysis of cell pellets containing non-soluble proteins after cell lysis demonstrated an over 40% increase in the PI content in FKBP2 KO compared to control cells (Figure 6D). These results point to a progressive accumulation of PI multimers with a subsequent loss of solubility due to impaired PI folding when cells lack functional FKBP2.

## 4. Discussion

The β-cell intrinsic mechanisms underlying deficient insulin secretion in T2D are still unclear, and much needs to be uncovered regarding PI biosynthesis, folding, and the role of ER protein chaperones in this process. PI folding, i.e., locking its 3D structure, is orchestrated in the ER, but apart from chaperoning by GRP94 and disulfide bridge formation, our insights into this process are limited [5,6,41]. This is in stark contrast to the recent identification of 38 PI-interacting proteins suggesting that multiple folding steps are required even before PI exits the ER and undergoes proteolytic cleavage to produce mature insulin [42]. Here, we discovered a novel role for FKBP2, a proline isomerase, in the folding of wild-type PI (Figure 6E). We found that FKBP2 physically interacts with PI and the PI chaperone, GRP94 and that the PI proline at position 28 is a likely target for FKBP2-dependent isomerization. Further, we demonstrated that β-cells with FKBP2 KO exhibit loss of intracellular PI associated with increased misfolding, loss of solubility, and increased PI secretion. Furthermore, FKBP2 KO cells showed increased apoptosis levels with no detectable changes in the expression of ER stress markers. Finally, we observed FKBP2 overexpression in β-cells from human islets obtained from T2D patients, likely as a compensatory response to the increased insulin biosynthetic demand.

In silico prediction models of the PI-FKBP2 interaction suggested that proinsulin interacts with FKBP2 through a set of conserved residues [43,44] involved in FKBP2’s binding of the inhibitor FK506, potentially necessary to facilitate proline isomerization (Figure 2D). This suggested to us that FKBP2 might be a bona fide PI isomerase. Human PI contains three proline residues, one in the B-chain and two in the C-peptide, with only P28 in the B-chain, being conserved among species (Figure 2E,F). Previous studies have shown that modifications of P28 have little to no effect on PI bioactivity, although they diminish PI dimer formation [45,46]. Somewhat contradictory to that, recently, a P28L mutation was identified in a maturity-onset diabetes of the young (MODY) patient, and molecular analysis identified partial defects in PI (P28L) oxidative folding and ER export [20]. The discrepancy may be explained by the fact that initial P28 modifications or substitutions were introduced outside of the physiological folding environment that is found in β-cells, whereas the P28L mutant undergoes folding attempts in vivo. In other words, determinants of foldability, i.e., P28, may not be apparent once the native protein state is reached by other means, i.e., in vitro oxidative PI folding [47,48]. If correct, it may suggest that P28 is indispensable for proper PI folding during the transition through ER. The ability of proline to adopt a cis or trans configuration accounts for proline’s tendency to bend the regional amino acid arrangement and disturb protein secondary structure by inhibiting an α-helix or β-sheet conformation, therefore allowing for subsequent protein folding [49,50]. In fact, FKBP2 binds more efficiently to unfolded PI than its folded state (Figure 6A), indicating that the potential proline isomerization takes place before the PI structure is locked by the formation of disulfide bonds. Finally, FKBP2 in an in vitro isomerization assay was efficient in the cis-to-trans isomerization of proline residue in a nanopeptide derived from the human PI sequence directly preceding and containing P28 residue (Figure 2G).

If FKBP2 works in vivo as a PI isomerase, i.e., converts cis isomers to trans, and if cis isomers disturb or even prevent PI folding, we would expect PI to misfold in FKBP2-deficient cells. We thus knocked out FKBP2 (Appendix A) and demonstrated a consistent phenotype of lower cellular contents of PI (at 2 and 20 mM glucose concentrations) and insulin (at 20 mM; Figure 3A). The PI content was partially restored after FKBP2 reconstitution in FKBP2 KO cells (Figure 3B).

The β-cell has at least two ways of dealing with misfolded PI: by an increase in PI secretion and via intracellular PI degradation. In fact, if PI is not processed within 1 h to mature insulin, it is directed to and secreted via the constitutive pathway [51], and an elevated PI secretion is indeed a hallmark of T2D and T1D [52,53,54]. PI secretion and intracellular degradation result in a shorter PI half-life, and in fact, this was our observation upon FKBP2 KO. As shown in Figure 5B, a shorter PI half-life in FKBP2 KO cells is partially a result of increased PI secretion, as prolongation of PI half-life was achieved via the treatment of cells with the ER–Golgi anterograde transport inhibitor, BFA. Importantly, FKBP2 KO impacts mostly PI folding status, as FKBP2-deficient cells maintain normal insulin secretion in response to glucose (Figure 5D).

The only partial restoration of PI content in secretion-inhibited FKBP2 KO cells may suggest that misfolded PI is actively degraded as well. As mentioned, FKBP2 has a preference for binding unfolded, fully reduced PI. This would place the P28 isomerization before the formation of PI disulfide bonds. As a consequence, increased formation of aberrant PI disulfide bonds would be anticipated. Indeed, FKBP2 KO cells contain relatively (to monomers) more PI multimers, including high-molecular-weight complexes (HMWC) that have been reported to represent PI disulfide-linked multimers (Figure 6B and Appendix A; [1]) where disulfide bonds are formed between different PI molecules. We supplemented this observation with the analysis of void SEC fractions, as large molecules (or their covalently linked complexes) have no or limited affinity to the SEC column resin pores and are thus eluted in the void fraction [55]. We found similar enrichment of PI disulfide-linked multimers in FKBP2 KO cells (Figure 6C upper blot) that could be reduced (Figure 6C lower blot), indicating that indeed these are multiple PI molecules linked via abnormal disulfide bonds. Furthermore, FKBP2 KO results in the accumulation of PI molecules in non-soluble cellular fractions (pellets, Figure 6D). This may be a result of two factors: 1. misfolded proteins, if not degraded promptly, tend to form aggregates, i.e., insoluble high-molecular-weight forms [56]; 2. it has been suggested that the flexibility of the PI B-chain C-terminus (residues B27-B30) contributes to the aggregation of insulin and formation of fibrils [57,58,59]. Consequently, modification to residues B27-B30 could lower PI solubility. Through the KO of FKBP2, we could potentially increase the presence of differentially spatially oriented and less soluble cis isomers. Taken together with the increase in HMWC, this may explain the accumulation of PI in the insoluble cellular fraction. The proof that HMWC indeed are formed predominantly by PI-containing P28 cis isomers could be provided through a direct detection of those isomers in PI molecules. Such a detection is potentially possible with tandem mass spectrometry combined with liquid chromatography, given appropriate controls and enzymatic digestion directed at trans or cis isomers are employed, and we are currently working on this methodology.

Furthermore, the above FKBP2 KO phenotype may explain why we did not observe changes to PI dimer formation (Figure 3C). It is plausible that only PI molecules containing trans P28 are competent to form dimers and they represent the majority of detectable PI, while cis P28 isomers are secreted and/or misfolded and form non-soluble aggregates (Figure 3C and Figure 6D).

It is important to note here that the observed phenotype where misfolded PI forms more HMWC and, at the same time, more PI is secreted may not represent the phenotype of FKBP2 KO or FKBP2 functional deficiency in normal β-cells. INS1-E cells used here secrete substantial amounts of PI while normal β-cells secrete only negligible amounts of PI [60,61]. Furthermore, observed increased PI secretion may be an adaptive feature of FKBP2 KO clonal cell lines used during our experiments. At the same time, FKBP2 functional deficiency in normal β-cells may lead exclusively to formation of PI HMWC, their accumulation in ER and induction of ER stress (discussed also below). Alternatively, one can propose that β-cells under duress employ multiple ways to deal with misfolded PI that include increased PI secretion and/or degradation, depending on, e.g., the time a given PI molecule spends in the ER awaiting further folding. To address those issues, FKBP2 functional deficiency via, e.g., pharmacological inhibition, needs to be induced in normal β-cells.

Models with PI misfolding in the ER display β-cell stress and apoptosis, insulin deficiency and the onset of diabetes [62]. In contrast, FKBP2 KO in β-cells led to the specific loss of PI and low-grade apoptosis, but no detectable ER stress induction or other detrimental effects were observed (Figure 4). How do we consolidate this induction of apoptosis without the activation of the ER stress pathways?

First, FKBP2 deficiency itself or through PI misfolding could potentially increase ER-to-mitochondria Ca^2+^ transport via the ER–mitochondria contact sites [63] and/or sustain an increase of cytosolic Ca^2+^, both inducing the intrinsic apoptotic pathway (reviewed in [64]), as is seen in prion-related disorders [65]. Second, the observed apoptosis may not be related to PI misfolding but rather represents a dysfunction of another FKBP2 target. In a recent survey of FKBP2-interacting proteins (data not shown), we identified subunits of mitochondrial ATP synthase that catalyzes ATP synthesis [66] or 60 kDa heat shock protein, HSPD1, which participates in the correct folding of proteins imported to mitochondria [67]. Interfering with the function of these proteins could induce the intrinsic mitochondrial apoptotic pathway without inducing ER stress [68]. However, it remains to be investigated if those proteins are functional FKBP2 targets. Third, during the generation of FKBP2 KO, cells that responded to KO and subsequent PI misfolding with ER stress followed by unresolved unfolded protein response might not have survived the clonal selection. Hence, the FKBP2 KO clones that survived and were used during this work might have undergone adaptation, the result of which is tuning down ER stress and low levels of apoptosis.

Taken together, these results indicate a functional link between FKBP2 and PI folding. Under conditions where an increased production of insulin is required (e.g., T2D), induction of FKBP2 expression may function as a compensatory response. Indeed, single-cell sequencing of pancreatic islets from human T2D donors demonstrated an up-regulation of FKBP2 mRNA in β-cells (Figure 1E); however, these data have not been independently validated (e.g., by qPCR).

## 5. Conclusions

In summary, our study has for the first time established the importance of FKBP2 as an essential PI isomerase in pancreatic β-cells and highlights a novel key function for PI P28 isomerization during PI folding. As such, our findings create an opportunity to investigate the potential of PI P28 isomerization status as a biomarker of early diabetes as well as a potential novel therapeutic target capable of improving PI folding.

## Figures and Tables

**Figure 1 biomolecules-13-00152-f001:**
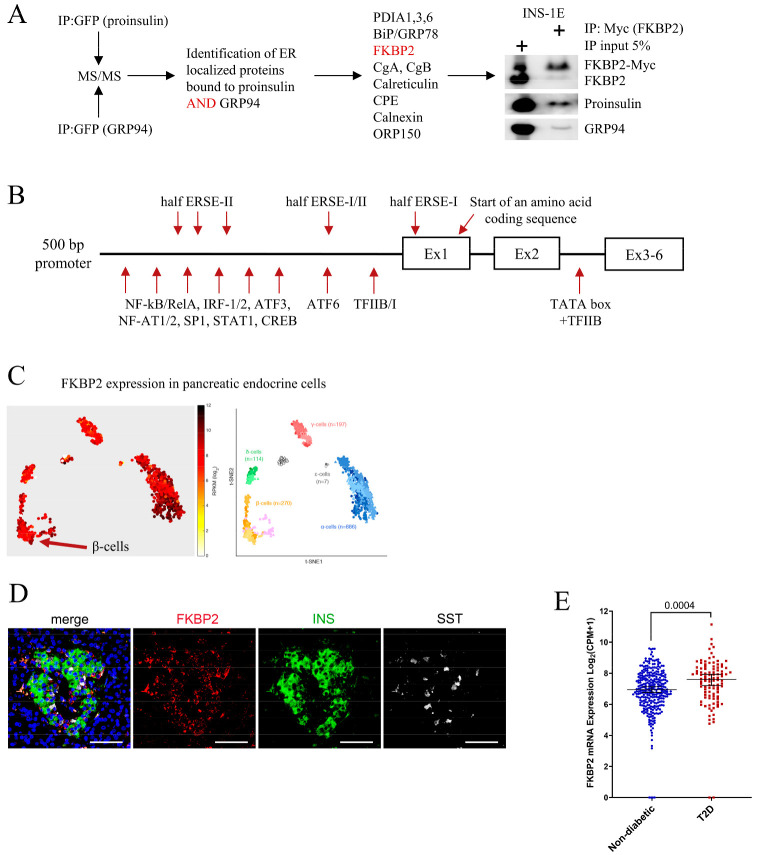
The FKBP2 isomerase interacts with proinsulin and GRP94 and is expressed in β-cells. (**A**). INS1-E cells were transiently transfected to express GFP-tagged human proinsulin or GRP94, both of which were immunoprecipitated and analyzed by mass spectrometry for binding partners present in the endoplasmic reticulum. FKBP2’s interaction with proinsulin and GRP94 was confirmed by SDS-PAGE/Western blot analysis of the immunoprecipitates of myc-tagged FKBP2 expressed in INS1-E cells. PDIA 1, 3, 6: Protein disulfide-isomerase 1, 3, 6; BiP/GRP78: binding immunoglobulin protein/Glucose Regulated Protein 78; CgA, CgB: chromogranin A and B; CPE: carboxypeptidase E; ORP150: oxygen-regulated protein 150. (**B**). Graphic representation of the FKBP2 promoter with putative transcription factor binding sites and regulatory elements identified using PROMO from ALGGEN. Analyzed DNA sequence was retrieved from HGNC:3718. ERSE: ER stress response element; TFIIB/I: transcription factor IIB/I; ATF6 and 3: activating transcription factor 6 and 3; NF-kB/RelA: nuclear factor-kB/REL-associated protein; IRF-1/2: interferon regulatory factor 1/2; NF-AT1/2: nuclear factor of activated T cells; SP1: Specificity Protein 1; STAT1: signal transducer and activator of transcription 1; CREB: cAMP response element-binding protein; Ex: exon. (**C**). FKBP2 single-cell gene expression. t-SNE representations colored according to FKBP2 expression levels, *n* = 1554 from Segerstolpe et al., Cell Metabolism 2016. (**D**). Representative fluorescence microscopy images of the human adult pancreas stained with antibodies against FKBP2 (red), INS (insulin, green) labeling β-cells and SST (Somatostatin, white) labeling δ-cells. Scale bars = 100 μm. DAPI (blue) labels the nuclei. (**E**). FKBP2 mRNA expression analyzed by single-cell RNA sequencing of β-cells derived from healthy and T2D individuals. Lines represent median with 95% CI error bars. Statistical analysis was performed using unpaired two-tailed parametric t-tests. FKBP2 transcript expression is shown as log2 (counts per million + 1).

**Figure 2 biomolecules-13-00152-f002:**
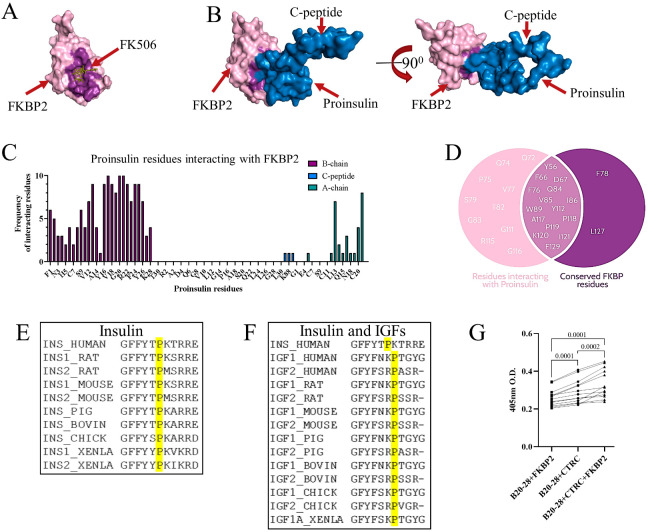
In silico modeling of proinsulin interaction with FKBP2, identification of proinsulin isomers, and demonstration of FKBP2 as a proinsulin isomerase. FKBP2 binding to its inhibitor FK506 ((**A**), crystal structure PDB: 4NNR) and proinsulin ((**B**), FKBP2 crystal structure PDB: 2PBC, proinsulin NMR structure PDB: 2KQP, blue) in the representative in silico models. In both structures, the residues involved in the interaction are highlighted in purple. The models were generated on the ZDOCK server. (**C**). The panel shows the frequency of proinsulin residues interacting with FKBP2 in the top ten generated interaction models between FKBP2 and proinsulin (2PBC and 2KQP structures, respectively). (**D**). Overlap between proinsulin interacting residues and conserved residues within FKBP2. The data displayed are the combined data from 100 FKBP2–proinsulin interaction models. The conserved residues are highlighted in dark purple. (**E**,**F**). Alignment of human proinsulin sequence (UniProt P01308), between positions 23 of the B-chain and position 1 of the C peptide with proinsulin (**E**) and IGF-1/2 (**F**) of Homo sapiens (HUMAN IGF1 P05019, IGF2 P01344), Rattus norvegicus (RAT INS1 P01322, INS2 P01323, IGF1 P08025, IGF2 P01346), Mus musculus (MOUSE INS1 P01325, INS2 P01326, IGF1 P05017, IGF2 P09535), Sus scrofa (PIG INS P01315, IGF1 P16545, IGF2 P23695), Bos Taurus (BOVIN INS P01317, IGF1 P07455, IGF2 P07456), Gallus gallus (CHICK INS P67970, IGF1 P18254, IGF2 P33717), and Xenopus laevis (XENLA INS1 P12706, INS2 P12707, IGF1A P16501). Sequences were retrieved from UniProt v. 2021_04. (**G**). Peptidyl-prolyl cis–trans isomerization assay of a peptide containing Pro28 (GERGFFYTP-F-pNA, B20–28) incubated with or without FKBP2. The substrate peptide was dissolved in LiCl/TFE buffer, and to initiate the reaction, chymotrypsin (CTRC) was added to digest and release the fluorophore (pNA) from the peptide in trans conformation. The absorbance was acquired at 405 nm. The signal from the time point with the highest absorbance in the reaction that contained FKBP2, n-Nitroanilide modified peptide and chymotrypsin was used to generate the graph. Data were analyzed by paired t-test between individually tested conditions.

**Figure 3 biomolecules-13-00152-f003:**
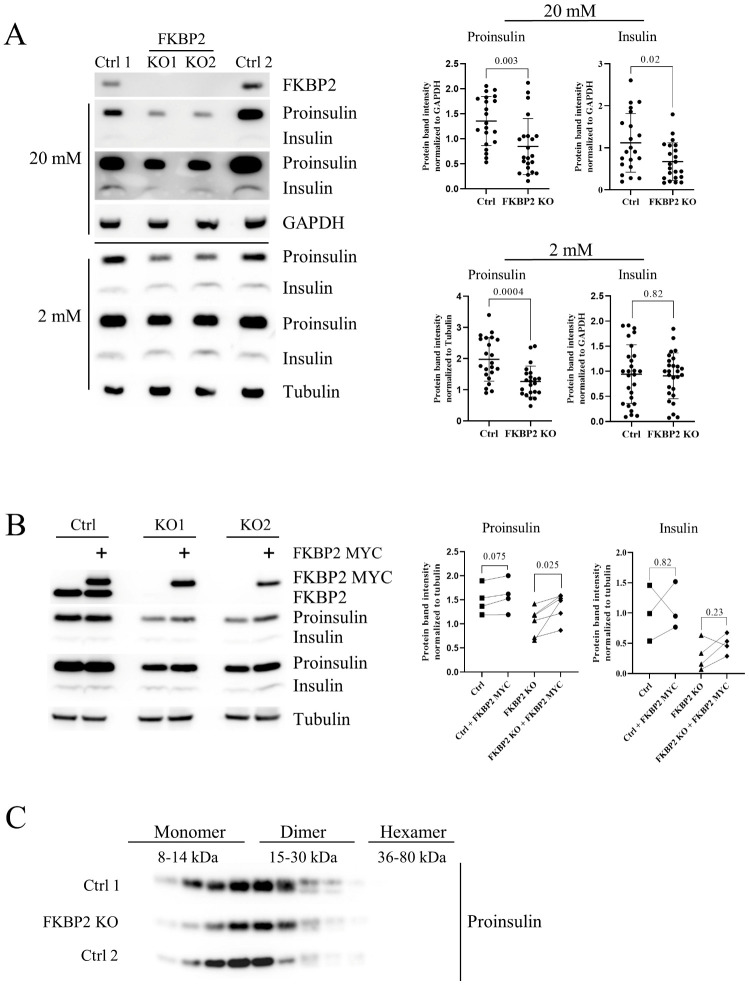
Diminished intracellular proinsulin and insulin contents after FKBP2 KO. (**A**). SDS-PAGE and Western blot analysis of proinsulin and insulin expression levels in FKBP2 KO and FKBP2 WT cells (failed FKBP2 KO and INS1-E) after 2 h in 20 or 2 mM glucose-containing media. FKBP2 KO was achieved via CRISPR/Cas9 guide FKBP2 directed-RNA targeting (clonal cell lines shown 3–6 months after viral transduction; representative blots of *n* > 10 on the left and band quantification on the right). Proinsulin and insulin were visualized with an anti-insulin antibody. (**B**). SDS-PAGE and Western blot analysis of proinsulin and insulin expression levels upon exogenous expression of FKBP2 in FKBP2 KO clones and INS1-E control (Ctrl) cells. Cells were transfected with plasmids coding for myc-tagged FKBP2, cultured for 48 h, and lysed 2 h after the introduction of fresh culture media supplemented with 20 mM glucose. Representative blot of *n* = 4 are presented on the left and band quantification on the right. (**C**). SDS-PAGE and Western blot analysis of proinsulin from FKBP2 KO, FKBP2 WT (Ctrl 1), and INS1-E (Ctrl 2) cells after size exclusion chromatography (SEC) separation at the pH 7.4 condition of cell lysis and SEC separation. SEC experiments *n* = 3. Data for A-B represent means ± SD analyzed by non-paired (**A**) or paired (**B**) t-test of treatments versus control.

**Figure 4 biomolecules-13-00152-f004:**
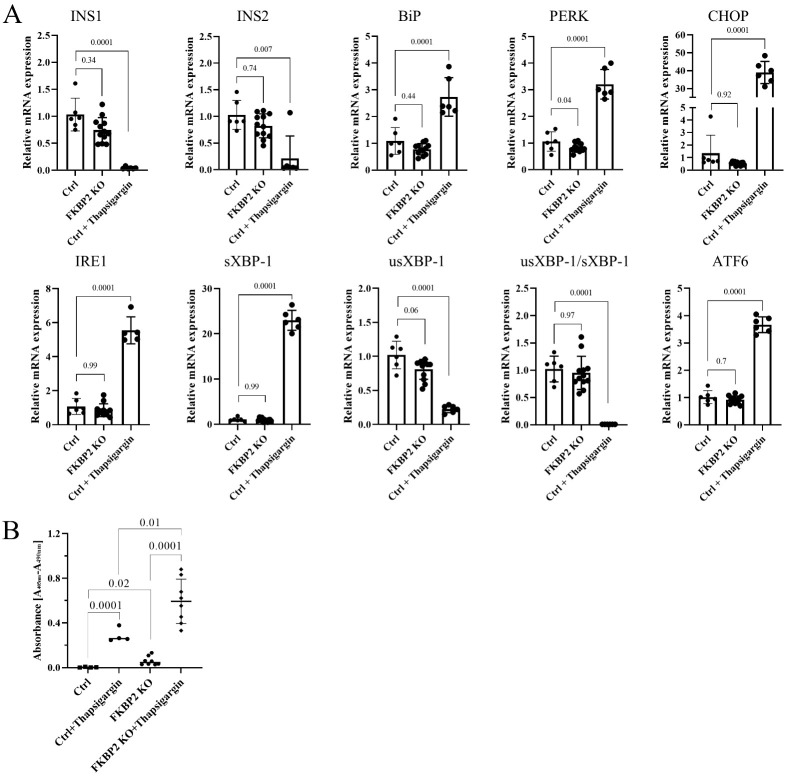
FKBP2 knockout does not induce endoplasmic reticulum stress but sensitizes cells to apoptosis. (**A**). mRNA levels of genes in the endoplasmic reticulum stress pathways were analyzed by quantitative reverse transcription-PCR (qRT-PCR) in Ctrl (FKBP2 WT cells) and FKBP2 KO cells. Data represent the mean ± SD analyzed by ordinary one-way ANOVA of treatments versus control, *n* > 6. (**B**). Apoptosis levels, representing internucleosomal degradation of genomic DNA, were analyzed in FKBP2 KO and Ctrl cells, *n* > 4. Where indicated, cells in A and B were exposed for 18 h to 0.7 µM of thapsigargin.

**Figure 5 biomolecules-13-00152-f005:**
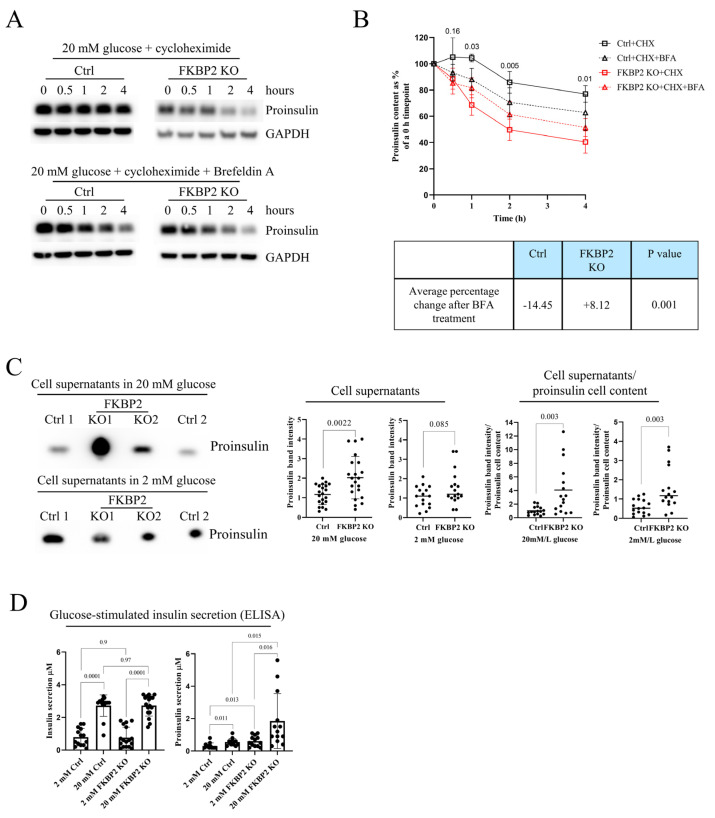
FKBP2 KO shortens proinsulin intracellular half-life and increases its secretion. (**A**,**B**). INS−1E control (Ctrl) and FKBP2 KO cells were cultured for 3 h in 2 mM glucose-containing media. subsequently, 100 µM of the protein synthesis inhibitor cycloheximide (CHX, upper panels) and 200 nM of the inhibitor of exocytosis Brefeldin A (BFA, bottom panels) were added to the culture media from the start of the experiment. Cells were lysed at indicated time points, analyzed via reducing SDS-PAGE, and proinsulin and insulin were visualized through Western blotting (WB) with an anti-proinsulin antibody, *n* = 5. B. WB data quantification. Top: graph *p* values were calculated by unpaired t-test of Ctrl + CHX vs. FKBP2 KO + CHX for each time point. Bottom: Table with averages of changes of Ctrl and FKBP2 KO cells over the course of experiment in proinsulin content (band intensity) in response to BFA treatment. (**C**). Accumulated secretion of proinsulin over the period of 4 h in 2 or 20 mM glucose by failed FKBP2 KO (Ctrl 1 and 2) and FKBP2 KO cells, analyzed by SDS-PAGE and WB (*n* = 10). In total, 500 µL of cell supernatants (adjusted to cell number) was concentrated using 10 kDa MWCO filters to remove salts and reduce the volume to 15 µL. Quantification of proinsulin bands was carried out with ImageJ (**A**,**B**) and normalized to GADPH (**A**) bands. Data are presented as means ± SD analyzed by unpaired t-test of treatments versus control. (**D**). The same cell types were tested for their ability to secrete insulin in response to the given glucose concentrations in KRBH buffer for a period of 30 min. Supernatants were analyzed by ELISAs specifically detecting mature insulin (left graph) or proinsulin (right graph) only. The bars represent the means ± SD.

**Figure 6 biomolecules-13-00152-f006:**
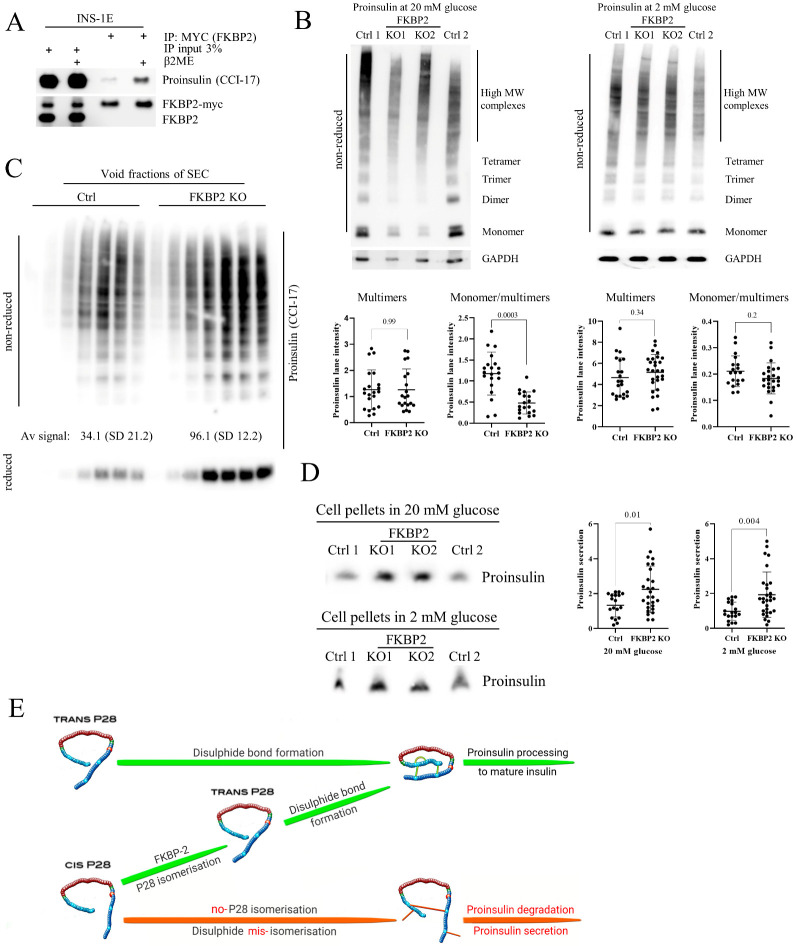
FKBP2 KO increases intracellular levels of high-molecular-weight proinsulin complexes and non-soluble proinsulin fraction. (**A**). INS-1E cells were transfected with myc-tagged FKBP2, followed 48 h later with immunoprecipitation (via myc tag) and SDS-PAGE/Western blotting to detect proinsulin. Where indicated, cell lysates were pretreated prior to immunoprecipitation for 10 min with 100 mM of reducing agent 2-mercaptoethanol (2-ME), *n* = 3. (**B**). Non-reducing SDS-PAGE/Western blot analysis of failed FKBP2 KO (Ctrl 1 and 2) and FKBP2 KO cells cultured for 3 h in 20 and 2 mM glucose conditions. The presence of high-molecular-weight proinsulin complexes was evaluated with a proinsulin-specific antibody as in Arunagiri et al. [1], *n* = 10. (**C**). Void fractions of size exclusion chromatography of failed FKBP2 KO (Ctrl) and FKBP2 KO cells (cultured for 3 h in 20 mM glucose-containing media) were analyzed under non-reducing and reducing conditions with SDS-PAGE/Western blot to detect proinsulin, *n* = 3. (**D**). Insoluble fractions of failed FKBP2 KO (Ctrl 1 and 2) and FKBP2 KO cells cultured for 3 h in 20 and 2 mM glucose conditions were treated with 2% SDS containing loading buffer, boiled for 20 min and analyzed by SDS-PAGE/Western blot to detect proinsulin, *n* = 10. Proinsulin detection was performed with the CCI-17 monoclonal antibody. All Western blot quantifications were carried out with ImageJ. The data were analyzed by the unpaired Student’s t-test and are presented as means SD ±. (**E**). A schematic representation of the proinsulin isomers’ folding steps is proposed in this work.

## Data Availability

Data are contained within the article or Appendix A.

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
