# Peer review of "FK506-Binding Protein 2 Participates in Proinsulin Folding"

_biomolecules, 2023, doi:10.3390/biom13010152_

Round 1
Reviewer 1 Report
In this paper, Hoefner et al has studied the importance of a proline isomerase FK506-binding protein 2 (FKBP2) on proinsulin folding. They studied this using Crispr-Cas in rat insulinoma cell line INS1E. They show that immature proinsulin secretion and proinsulin high molecular weight complexes are increased in knock out cells. They also show increased FKBP2 mRNA in T2D cases and suggest that it is a compensatory response.
This paper will require minor revisions.
Can overexpression of FKBP2 in FKBP2 knock-out cells decrease high molecular weight proinsulin complexes and non-soluble proinsulin fraction?
Figure 5C: Show cell supernatant per proinsulin content in cells.
The Supp. Figures 1 and 3 are missing. There are two Supp. Figures 2 and 4, but they do not have labels or Figure text. Please add.
Discussion lane 491: “FKBP2 KO cells contain relatively more PI multimers...Fig 6B, and Suppl. Fig. 4”.
In Figure 6A multimers p=0.99, not significant. Suppl. Fig. 4 “AMS treatment” does not have labels. Please add arrows or line to show multimers.
Discussion lane 509: “... we did not observe changes to PI dimer formation (Fig. 3A).” There are no dimers shown in Figure 3A.
References: 1 and 42 is the same.
Author Response
We wanted to thank the Reviewer for the input and questions that directed us to focus and clarify but also to enrich our manuscript. Our answers are marked in italic.
Specific questions or issues raised:
- Can overexpression of FKBP2 in FKBP2 knock-out cells decrease high molecular weight proinsulin complexes and non-soluble proinsulin fraction?
The Reviewer’s question points to the right direction that would not only further underlie FKBP2 as a bona fide protein responsible for observed phenotype but also indicate potential therapeutic approach. We have discussed this experiment during our internal meetings, when the issue of FKBP2 reexpression on FKBP2 KO background was proposed. Using the same lysates from experiments presented in figure 3B, we have run non-reducing SDS-PAGE and Western blotting to detect changes in the presence of high molecular weight proinsulin complexes (HMWC). However given only partial reconstitution of proinsulin under those experimental conditions (reducing SDS-PAGE), we were unable to detect a decrease in HMWC. The reasons for this might be multiple, including clonal adaptation combined with exogenous expression of FKBP2 and/or lower sensitivity of non-reducing SDS-PAGE. As reconstitution of proinsulin content upon FKBP2 reexpression, despite its all important implications, was not a focus of this publication, we did not pursue the issue further. At this point, we have no personnel resources to perform additional experiments as our laboratory will be closed in the beginning of 2023.
- Figure 5C: Show cell supernatant per proinsulin content in cells.
Graphs showing cell supernatant per proinsulin content in cells have been added.
- The Supp. Figures 1 and 3 are missing. There are two Supp. Figures 2 and 4, but they do not have labels or Figure text. Please add.
Corrected and added.
- Discussion lane 491: “FKBP2 KO cells contain relatively more PI multimers...Fig 6B, and Suppl. Fig. 4”.
In Figure 6A multimers p=0.99, not significant.
We have corrected the sentence to indicate that there more multimers relatively to monomers in FKBP2 KO vs Ctrl cells.
Suppl. Fig. 4 “AMS treatment” does not have labels. Please add arrows or line to show multimers.
The figure has been corrected.
- Discussion lane 509: “... we did not observe changes to PI dimer formation (Fig. 3A).” There are no dimers shown in Figure 3A.
The correct reference should be Fig. 3C representing SEC analysis of samples, where monomer, dimer and hexamer are separated according to their weight before analyzing on SDS-PAGE. We amended the manuscript text.
- References: 1 and 42 is the same.
Reference 42 has been removed and corrected to 1.
Yours sincerely,
Michal Tomasz Marzec
Associate Professor
Reviewer 2 Report
Title: FK506-binding protein participates in proinsulin folding Comments: Line 72: amino acid “swap”. Correct it in other places as well. Line 115: Donor number 2 with a high BMI and elevated HbA1c levels is strongly indicative of prediabetes. There is evidence to suggest that prediabetic beta-cell morphology and behaviour is markedly different from wild-type beta cell, with a higher basal insulin secretion and early signs of protein misfolding. Glucose Stimulation: To account for first phase and second phase, glucose stimulation is carried out typically for about two hours. The phrasing used in this paper states the stimulation to be four hours but perhaps the authors have also included the 2 hour baseline incubation in these four hours as well, which is, in my opinion, redundant. Section 2.3. It would be worthwhile to include the information about T2D patients’ pancreatic sections here from the supplementary. Figure 1C and 1D: some issues with the panel quality, increase DPI if possible. Figure 1E: Sample size of non-T2D and T2D is not the same but since the p-value is small, this can be let go. Figure 2: Highlight A, B, C, D in the caption. Conclusion: Manuscript approved after these minor formatting revisions.
Author Response
We wanted to thank the Reviewer for the input and questions that helped us clarify and enrich our manuscript. Our answers are marked in italic.
Reviewer’s comments:
Line 72: amino acid “swap”. Correct it in other places as well.
Corrected.
Line 115: Donor number 2 with a high BMI and elevated HbA1c levels is strongly indicative of prediabetes. There is evidence to suggest that prediabetic beta-cell morphology and behaviour is markedly different from wild-type beta cell, with a higher basal insulin secretion and early signs of protein misfolding.
We agree with the Reviewers observation, however, as noted in the manuscript text, those donors are non-diabetic (as per clinical diagnosis), which does not preclude the possibility of their prediabetic state. It is possible that beta-cell physiology is already affected on varying levels, however, we have observed no differences in staining for insulin and FKBP2 between three tested samples. Particularly, similar insulin staining suggests that beta-cells in question are in, or close to, typical wild-type beta cells conditions (as judged by the sample comparison among all three donors).
Nevertheless, we have added a text indicating the Reviewers concern re. Donor 2, as follows: this donor characteristics might be considered prediabetic.
Glucose Stimulation: To account for first phase and second phase, glucose stimulation is carried out typically for about two hours. The phrasing used in this paper states the stimulation to be four hours but perhaps the authors have also included the 2 hour baseline incubation in these four hours as well, which is, in my opinion, redundant.
Fig. 5C shows accumulated proinsulin secretion over the period of 4 hours in the 2 or 20 mM glucose concentrations. This has been done to achieve greater proinsulin concentration in the supernatants, enabling us to perform SDS-PAGE and Western blot analysis.
Fig. 5D on the other hand shows a fairly typical glucose-stimulated insulin secretion (GSIS) where the 2 hour baseline incubation precedes 1 hour incubation in 2 and then 20 mM glucose concentrations.
Section 2.3. It would be worthwhile to include the information about T2D patients’ pancreatic sections here from the supplementary.
Section 2.3 has been expanded.
Figure 1C and 1D: some issues with the panel quality, increase DPI if possible.
We have reformatted Fig. 1 to increase quality.
Figure 2: Highlight A, B, C, D in the caption.
A-F highlighted.
Yours sincerely,
Michal Tomasz Marzec
Associate Professor
Reviewer 3 Report
The original research article “FK506-binding protein participates in proinsulin folding” has demonstrated that FKBP2 contributes to proinsulin folding. The authors have shown that FKBP2 inhibition promoted proinsulin turnover with reduced intracellular proinsulin and insulin levels. These results were supported by an incased proinsulin secretion and formation of proinsulin high molecular weight complexes. In addition, it was shown that FKBP2 inhibition increased apoptosis without altering ER-stress response genes. Finally, the authors have also suggested that proline at position 28 of the proinsulin B chain is the substrate of FKBP2 isomerization activity. Overall, in my opinion, this is a well written report and experimental concept is novel and the results are solid. I don’t have any specific concerns and questions.
1. Please correct the title to FK506-binding protein 2
Author Response
We wanted to thank the Reviewer for spotting the omission in the title and evaluating positively our manuscript.
The title of the manuscript has been corrected.
Yours sincerely,
Michal Tomasz Marzec
Associate Professor